# Development of a Novel, Ecologically Friendly Generation of pH-Responsive Alginate Nanosensors: Synthesis, Calibration, and Characterisation

**DOI:** 10.3390/s23208453

**Published:** 2023-10-13

**Authors:** Abdalaziz Alwraikat, Abdolelah Jaradat, Saeed M. Marji, Mohammad F. Bayan, Esra’a Alomari, Abdallah Y. Naser, Mohammad H. Alyami

**Affiliations:** 1Department of Applied Pharmaceutical Sciences and Clinical Pharmacy, Faculty of Pharmacy, Isra University, P.O. Box 33, Amman 11622, Jordan; ad1462@iu.edu.jo (A.A.); abdolelah.jaradat@iu.edu.jo (A.J.); esraa.alomari@iu.edu.jo (E.A.); abdallah.naser@iu.edu.jo (A.Y.N.); 2Faculty of Pharmacy, Philadelphia University, P.O. Box 1, Amman 19392, Jordan; smarji@philadelphia.edu.jo (S.M.M.); mbayan@philadelphia.edu.jo (M.F.B.); 3Department of Pharmaceutics, College of Pharmacy, Najran University, Najran 66462, Saudi Arabia

**Keywords:** pH nanosensors, ecologically friendly, MHDs, PDI, fluorophore conjugation, fluorescence intensity

## Abstract

Measurement of the intracellular pH is particularly crucial for the detection of numerous diseases, such as carcinomas, that are characterised by a low intracellular pH. Therefore, pH-responsive nanosensors have been developed by many researchers due to their ability to non-invasively detect minor changes in the pH of many biological systems without causing significant biological damage. However, the existing pH-sensitive nanosensors, such as the polyacrylamide, silica, and quantum dots-based nanosensors, require large quantities of organic solvents that could cause detrimental damage to the ecosystem. As a result, this research is aimed at developing a new generation of pH-responsive nanosensors comprising alginate natural polymers and pH-sensitive fluorophores using an organic, solvent-free, and ecologically friendly method. Herein, we successfully synthesised different models of pH-responsive alginate nanoparticles by varying the method of fluorophore conjugation. The synthesised pH nanosensors demonstrated a low MHD with a relatively acceptable PDI when using the lowest concentration of the cross-linker Ca^+2^ (1.25 mM). All the pH nanosensors showed negative zeta potential values, attributed to the free carboxylate groups surrounding the nanoparticles’ surfaces, which support the colloidal stability of the nanosensors. The synthesised models of pH nanosensors displayed a high pH-responsiveness with various correlations between the pH measurements and the nanosensors’ fluorescence signal. In summation, pH-responsive alginate nanosensors produced using organic, solvent-free, green technology could be harnessed as potential diagnostics for the intracellular and extracellular pH measurements of various biological systems.

## 1. Introduction

Nanotechnology has long been seen as a paradigm for carrying drug therapeutics due to the several advantages that the nanoparticles provide, spanning from a prolonged circulation, enhanced drug efficacy/potency, lowered side effects, and site-specific drug delivery [1].

Biosensors have been recently developed for diagnostic purposes and for the treatment of many diseases [2,3]. For instance, it has been reported that multiple kinds of biosensors were utilised for early detection and diagnosis of prostate cancer to enhance clinical intervention and treatment outcomes. These biosensors were based on either optical or electrochemical–magnetic methods to detect prostate tumour biomarkers such as the prostate-specific antigen [4]. Additionally, recent advances in designing integrated detection devices based on biochemical processes combined with electrical control to promote non-invasive and more convenient diagnostics have also been extensively addressed [5].

More specifically, the biosensors involved in the measurement of intracellular or extracellular pH that are used for disease diagnosis have attracted the attention of many researchers in comparison to traditional sensing methods, e.g., fibre optics, and electrodes, that are invasive to cells and can cause alterations to cellular functions [6,7,8]. Accordingly, many pH-responsive fluorophores are considered alternatives to conventional pH measuring techniques due to their fast, non-invasive, and real-time measurement capabilities of cytosolic pH in living cells [9,10]. Moreover, using multiple pH-sensitive fluorophores leads to an increase in the brightness of the system that could enhance the signal-to-noise ratio, thus enabling augmented sensitivity for pH detection [11]. In light of this, new pH-sensing methods that involve the incorporation of multiple fluorescent dyes have been discovered, thus avoiding cellular damage by protecting the cellular microclimate from direct exposure to those toxic fluorescent molecules [12,13,14].

Cellular pH is a critical parameter that could affect many biochemical and metabolic cellular pathways. Therefore, a plethora of research studies have been conducted in order to detect the extracellular and intracellular pH of mammalian cells and/or organisms in vivo [15,16,17,18,19]. For example, it has been reported that polyacrylamide nanosensors were utilised for real-time measurements of the intracellular pH changes of yeast cells during glucose metabolism [20]. Moreover, it has been reported that silica nanosensors consisting of pH-sensitive fluorescein isothiocyanate demonstrated a capability for monitoring cellular processes occurring in the cytosol (pH∼7) and in slightly acidic vesicles such as the lysosomes and the endosomes (pH∼5) [21].

The pH detection range was further extended by incorporating an additional pH-sensitive fluorophore that is responsive to a lower pH range (3.5 to 5), such as the Oregon green, which was incorporated into an inert polyacrylamide matrix. The produced system successfully detected the pH values in an extended range (3.5–7.5) [22]. The pH nanosensors with an extended dynamic range had significant in vivo applications and were utilised for the measurement of the pH changes throughout the pharyngeal and intestinal lumen inside C. elegans worms [23].

Despite a myriad of techniques and pH-responsive nanosensors exploited for detecting the changes in the intracellular and extracellular pH, the safety profile of these nanosensors after cellular delivery is not well addressed. This could be attributed to the fact that most these systems rely on silica nanoparticles that might induce immunotoxicity [24], polyacrylamide nanoparticles that are composed of repeating monomers which could increase the cellular apoptosis/cytotoxicity [25], or quantum dots that could lead to cellular toxicity [26]. Furthermore, safety concerns pertaining to acrylamide residuals have been reported, which may range from 0.01 to 0.1% after the polymerisation of polyacrylamide nanoparticles. These concentration levels could lead to reproductive system impairments in males as well as to the induction of neurotoxicity brought about by DNA damage [27].

Moreover, there are tremendous environmental concerns about synthesising these nanosensors as they require using large quantities of organic solvents for the production and/or washing processes. For example, silica nanoparticles require ethanol as a reaction solvent during the synthetic procedure and large quantities of ethanol for the washing process [28,29,30]. It has been reported that ethanol spillage in the environment causes detrimental damage to the aquatic ecosystem by displacing the oxygen dissolved in the water [31,32].

Similarly, the production of polyacrylamide nanoparticles requires hexane as a solvent for the polymerisation process, as well as a thorough washing using ethanol [33,34,35]. It has been reported that a high concentration of hexane can lead to acute toxicity in both humans and animals, whilst a continuous exposure to low levels of hexane may cause neurotoxicity in living organisms [36,37,38].

Moreover, the synthetic procedure of quantum dots (QDs) such as CdSe QDs also requires chloroform as a solvent, acetone as a non-solvent, and tributyl phosphine to dissolve selenium [39,40,41,42,43]. It has been shown that the discharge of chloroform into the environment causes acute toxicity in aquatic life, including various marine species [44,45].

Therefore, there is an ecological need to formulate new nanosensors that are biocompatible, biodegradable, environmentally safe, and ecologically friendly. Accordingly, this study aimed to design novel, safe, and eco-friendly pH-responsive nanosensors consisting of pH-sensitive fluorophores covalently attached to an alginate polysaccharide, which is a non-toxic, biodegradable, biocompatible, and natural polymer, using the ionic gelation method [46,47,48,49], which is an organic, solvent-free method that primarily relies on using a divalent cation as the cross-linking agent, thus producing such a new generation of eco-friendly pH nanosensors, which could be safely employed in biosensing applications in the future, eliminating any environmental hazards imposed by the existing pH nanosensors.

## 2. Materials, Equipment, and Methods

### Materials and Equipment

Alginic acid sodium salt low viscosity (ALG), 6-aminofluorescein, MES hydrate, and sodium periodate were all supplied by Sigma-Aldrich, St. Louis, MO, USA. Glacial acetic acid and sodium hydroxide were supplied by Az Chem, Hangzhou, China, whereas calcium chloride and disodium hydrogen phosphate were supplied by Ghtech, Guangzhou, China. N,N-dimethyl formamide (DMF) was supplied by Tedia, Fairfield, OH, USA, and sodium cyanoborohydride was supplied by Santa Cruz Biotechnology, Dallas, TX, USA. Hydrochloric acid (HCl) 33–36% was supplied by BBC Chemicals, Lucknow, India, and 1-(3-dimethylaminopropyl)-3-ethylcarbodiimide (EDC) was sourced from Biosynth Carbosynth, Compton, UK. Ammonia solution 32%, citric acid, and ethane diol were supplied by Alpha Chemika, Andheri, India, while sodium acetate trihydrate was supplied by Central Drug House Ltd., Delhi, India. Phosphate buffered saline (PBS) and Oregon Green^®^ 488 cadaverine-5-isomer were supplied by Thermo Fisher Scientific, Waltham, MA, USA. Except when otherwise noted, all of the substances used in this experiment were of analytical grade.

Size, PDI, as well as zeta potential were assessed using the Malvern Panalytical Zetasizer Ultra. The PerkinElmer Spectrum Two FT-IR Spectrometer was used to present the FTIR spectra. The SnakeSkin dialysis tubing was made available by Thermo Fisher Scientific, USA. An American microfluidic pump model No. 300 and a British dolomite microfluidic chip made up the system utilized to create the nanoparticles. The Microfluidic Chip was observed using a Nikon Alphaphot YS lens from Nikon Corporation, Tokyo, Japan, and fluorescence was measured using a FLx800 by Biotek Instrument Inc., Winooski, VT, USA. The Sonorex Digitec from BANDELIN electronic, Berlin, Germany, was utilized for sonication, while the Heto FD1.0 from Heto-Holten, Allerød, Denmark, was used for freeze-drying. Additionally, the Labofuge 200 from Heraeus Sepatech, Hanau, Germany, was employed. All the equipment was cleaned or replaced after each use, as appropriate, and all the devices were maintained regularly.

## 3. Methods

### 3.1. Synthesis of Fluorescein- and Oregon Green-Labelled Alginate Polymers by End-Chain and Intra-Chain Conjugation of the Fluorophore

For all the pH-sensitive-nanosensors-based models, fluoresceinamine was attached to the end of the sodium alginate chain (the reducing end) using sodium cyanoborohydride to produce fluorescein-labelled alginate, as illustrated in Figure 1, which was synthesised as follows: 100 mg of alginate was dissolved in 10 mL of ammonium acetate with pH 5 and stirred until it was completely dissolved. Then, 4 mg of 6-Aminofluorescein was dissolved in a separate flask containing 1 mL of N,N Dimethyl-formamide (DMF) and 10 mL of ammonium acetate having a pH of 5. After that, 30 mg of sodium cyanoborohydride was added to a round bottom glass containing the above solutions to reduce the formed iminium intermediate. After that it was stirred for 4 days at 65 °C. Fluorescein-labelled polymer samples were dialysed with water as a medium using a 3.5 kDa dialysis membrane for 72 h, followed by freeze-drying for 48 h, and were stored in the refrigerator; all the samples were kept under dark conditions.

For the pH-sensitive nanosensors prepared using model I, Oregon green was attached to the end of the sodium alginate chain (the reducing end) using sodium cyanoborohydride in order to produce Oregon green-labelled alginate, as displayed in Figure 2, which was synthesised as follows: 50 mg of alginate was dissolved in a 10 mL of ammonium acetate buffer with pH 5 and stirred until completely dissolved. In a separate flask containing 2 mL of ammonium acetate buffer with pH 5 and 100 μL DMF, 0.2 mg of Oregon green cadaverine was dissolved. Then, 30 mg of sodium cyanoborohydride was added to a round bottom glass containing the above solutions to reduce the formed iminium intermediate. After that, the mixture was stirred for 4 days at 65 °C. Then, Oregon green-labelled polymer samples were dialysed with water as a medium using a 3.5 kDa dialysis membrane for 72 h, followed by freeze-drying for 48 h, and were stored in the refrigerator. Finally, all the samples were wrapped with aluminium foil and kept in a dark condition.

For the pH-sensitive nanosensors prepared using model II, the Oregon green was attached within the chain of alginate polymers using a 1-(3-Dimethylaminopropyl)-3-ethylcarbodiimide (EDC) coupling reaction to produce Oregon green-labelled alginate, as demonstrated in Figure 2, which was synthesised as follows: 50 mg of alginate was dissolved in a solution containing 50 mL of MES buffer with pH 5.6 and 3.5 μL EDC. The pH of the above solution was lowered to 1.5 using HCl in order to enhance the solubility of the base form of EDC. In a separate flask, a solution containing 1 mg/mL of Oregon green cadaverine was prepared by dissolving 0.115 mg of the fluorophore in 115 μL of medium containing an equal volume ratio of DMF and MES buffer. Immediately, that formed a cloudy solution upon addition. After mixing these solutions, the pH was raised to 5.6, which is the optimum pH for the EDC coupling reaction (Figure 3), using NaOH and was stirred for 24 h. Oregon green-labelled polymer samples were dialysed with water as a medium using a 3.5 kDa dialysis membrane for 72 h, followed by freeze-drying for 48 h, and were stored in the refrigerator. Then, all the samples were kept with aluminium foil in dark conditions.

For the pH-sensitive nanosensors prepared using model III and IV, the Oregon green was covalently incorporated into two oxidised forms of the alginate chain, with 10% and 80% oxidation percentages; this was achieved using a specified amounts of either 11 mg or 63 mg of sodium periodate, as illustrated in Figure 4A,B, respectively. The fluorophore attachment was attained through the reductive amination process of the aldehyde groups formed within the polymer chain to produce Oregon green-labelled alginate, which was synthesised as follows: 100 mg of alginate and specified amounts of sodium periodate (11 mg for model III and 63 mg for model IV) were dissolved in a 5 mL sodium acetate buffer with pH 5 and stirred for 24 h at 4 °C in the dark, followed by the addition of 1 mL of ethylene glycol to quench the excess, unreacted periodate. The solutions were dialysed with water as a medium using a 3.5 kDa dialysis membrane for 24 h in order to produce the two oxidised forms of alginate (10% and 80%), followed by freeze-drying for 48 h, and were stored in the refrigerator. Later, 50 mg of each oxidised form of alginate was dissolved in 10 mL of acetate buffer and stirred until completely dissolved. Then, 0.2 mg of Oregon green was dissolved in a separate flask containing a mixture of 1 mL DMF and 2 mL ammonium acetate having a pH of 5. Then, 30 mg of sodium cyanoborohydride was added to a round bottom glass containing the above solutions to reduce the formed iminium intermediate. After that, both solutions were stirred for 4 days at 65 °C. The Oregon green polymer samples were dialysed with water as a medium using a 3.5 kDa dialysis membrane for 72 h, followed by freeze-drying for 48 h, and were stored in the refrigerator. Finally, all the samples were kept in the dark until further use.

### 3.2. Synthesis of pH-Sensitive Alginate Nanosensors Using Microfluidic Technology

A microfluidic system comprised of a microfluidic chip and flow-controlled syringe pumps was utilised to produce alginate-based pH nanosensors. A hydrodynamic flow-focusing chip (X-type microfluidic chip) was connected to three inlet lines constituting two lateral lines and one central line. Model I was prepared by mixing equal volumes of fluorescein-labelled alginate polymers and Oregon green-labelled alginate polymers with both fluorophores attached to the end of the alginate chain, as demonstrated in Figure 1 and Figure 2, respectively. In model II, pH nanosensors were prepared by mixing equal volumes of fluoresceinamine-labelled alginate with the fluorophore attached to the end of the chain, as displayed in Figure 1, and Oregon green-labelled alginate with the fluorophore attached within the chain of alginate using an EDC coupling reaction, as displayed in Figure 3. In model III, pH nanosensors were prepared by mixing equal volumes of fluorescein-labelled alginate polymers with the fluorophore attached to the end of the alginate chain, as indicated in Figure 1, and Oregon green-labelled alginate with the fluorophore attached within the polymer chain of oxidised alginate (10% oxidation), as indicated in Figure 4A. In model IV, pH nanosensors were prepared by mixing equal volumes of fluorescein-labelled alginate polymer with the fluorophore attached to the end of the alginate chain, as indicated in Figure 1, and Oregon green-labelled alginate with the fluorophore attached within the polymer chain of oxidised alginate (10% oxidation), as indicated in Figure 4B. The mixture of fluorescein- and Oregon green-labelled alginates was introduced in the central inlet line of the chip at a concentration of 0.4 g/100 mL (0.4% (*w*/*v*), which was run at a flow rate of 0.25 mL/min. On the other hand, different molar concentrations of the calcium cross-linker were prepared (1.25, 2.5, 5, and 10 mM), subsequently introduced in the later inlet lines of the chip, and run at a specific rate of 0.5 mL/min to enable mixing and crosslinking with the fluorescently labelled alginate chains in the central line. The resultant pH nanosensors were collected after 2 min, centrifuged, and washed with distilled water three times at 5200 rcf (g). The pH nanosensors were covered with aluminium foil and stored in the fridge for further use.

### 3.3. Alginate Nanosensors’ Calibration Using a Fluorescence Microplate Reader

Buffers with different pH values ranging from 3.5 to 8.0 were prepared by adding the specified volumes of sodium phosphate buffer (0.2 M) to specific volumes of citric acid buffer (0.1 M). Then, the synthesised pH nanosensors were suspended in these buffers to obtain a constant concentration of 1 mg/mL for the pH nanosensors suspended in buffers with pH values of 3.5, 4, 4.5, 5, 5.5, 6, 6.5, 7, and 7.5. The prepared models of pH nanosensors suspended in the above-mentioned buffers with various pH values were all placed in a 96-well plate; then, the fluorescence intensity for each sample was recorded at an emission wavelength of 538 nm using an excitation wavelength of 485 nm through a fluorescence microplate reader. The acquisition settings for the fluorescence microplate reader were as follows: a sensitivity of 40 and a slit width of 20 nm to obtain the brightest fluorescence intensity. The fluorescence intensities for each pH-responsive model suspended in each well (i.e., each buffer with a specific pH value) were plotted against each pH value to generate the pH-fluorescence response curves of the produced pH nanosensors.

## 4. Results and Discussion

### 4.1. Characterisation of the Fluorophore-Tagged Alginates and the Designed pH-Responsive Alginate-Based Nanosensors

#### 4.1.1. Confirmation of Fluorescence Labelling of Alginate with pH-Sensitive Fluorophore Using FTIR

Figure 5 illustrates the FTIR spectra for the native alginate polymer (ALG-LMWT) alone, the fluoresceinamine alone, the Oregon green cadaverine (OG-CAD) alone, the fluoresceinamine-labelled alginate (FLU-ALG) prepared in Figure 1, and the Oregon green labelled alginate with the fluorophore conjugated to the alginate chain using either the direct reductive amination (OG-ALG (RA)) prepared in Figure 2 or the 1-Ethyl-3-(3-dimethylaminopropyl) carbodiimide (EDC) coupling reaction (OG-ALG (EDC)) prepared in Figure 3. The FTIR spectrum of the native alginate (ALG (LMWT)) demonstrated a characteristic absorption peak at around 1600 cm^−1^ related to the carbonyl stretching of the alginate’s carboxylate groups. In comparison, the OG-ALG (RA))’s spectrum demonstrated a characteristic peak at around 1730 cm^−1^, in addition to the 1600 cm^−1^ absorption peak displayed by the alginate’s carboxylates. This could be attributed to the presence of the lactone’s or cyclohexanone’s carbonyl group of the attached Oregon green, indicating a successful end-chain conjugation of the Oregon green to the alginate’s reducing end via the reductive amination process. Moreover, the OG-ALG (RA))’s spectrum showed a characteristic absorption peak at 1250–1290 cm^−1^ that represents the C–N bond stretching, indicating the presence of the amine groups of the Oregon cadaverine after their successful attachment to the native alginate that does not contain amine groups. Similarly, (Flu-ALG) demonstrated an absorption peak at around 1700 cm^−1^ that represents the stretching of the carbonyl group of the lactone ring or the cyclohexanone of the fluorescein, in addition to the absorption peak at 1600 cm^−1^ brought about by the native alginate’s carboxylate groups. This indicates the successful conjunction of the fluorescein to the alginate. This conjugation was also confirmed by the presence of an additional 1250–1290 cm^−1^ representing the stretching of the C–N bond, indicating the presence of the amine groups of the fluorescein attached to the native alginate that intrinsically precludes any amine groups. On the other hand, the FTIR spectrum of the OG-ALG (EDC) demonstrated a characteristic absorption peak at 1650 cm^−1^, in addition to the native alginate’s carboxylate groups’ carbonyl peak (1600 cm^−1^). This sharp peak (1650 cm^−1^) represents the carbonyl stretching in the amide bond that is expected to form between the alginate’s carboxylate and the amine group of the Oregon green during the EDC coupling reaction. This reinforces the successful intra-chain conjugation of the Oregon green within the alginate chain via the EDC reaction. Additional evidence of the Oregon green’s conjugation is the appearance of an absorption peak at around 1730 cm^−1^, which represents the stretching of the carbonyl group of the lactone ring or the cyclohexanone in the Oregon green that is not present in the alginate.

Moreover, FTIR spectra were plotted (Figure 6) for the oxidised forms of the native alginate polymers after chemical modification using periodate oxidation with either 10% oxidation (ALG (10% PER)) or 80% oxidation (ALG (80% PER)), as well as for the oxidised alginate polymers (10% and 80% periodate oxidation) covalently labelled with the Oregon green cadaverine (OG-CAD) through the reductive amination process, as previously described in Figure 4A,B. It can be observed that both oxidised forms of alginate at 10% and 80% oxidation demonstrated an absorption peak at around 1730 cm^−1^, representing the carbonyl (C=O) stretching of the oxidised alginate’s aldehyde groups, in addition to the absorption peak of the carboxylate group (1600 cm^−1^) present in the native alginate. This indicates the successful oxidation of the alginate chain that results in the glycol splitting of the glucuronic acid residues and the formation of aldehyde groups within the chain of alginate polymer. On the other hand, the FTIR spectra of the Oregon green-labelled oxidised alginate polymers demonstrated a peak at 1730 cm^−1^. However, the absorption intensity of this peak that corresponds to the aldehyde groups was slightly lower than that observed for the oxidised alginate polymers before the Oregon green’s conjugation. This could also give a clue that some of the aldehyde groups in the oxidised alginate were converted to secondary amine groups upon the Oregon green’s chemical conjugation during the reductive amination process that is illustrated in Figure 4A,B. Overall, Table 1 summarises all the IR absorption peaks along with the functional groups in each alginate polymer before and after dye conjugation with various intra-chain and end-chain fluorophore attachment processes.

In summation, although the FTIR results demonstrated the successful conjugation of both the fluorescein and the Oregon green to the alginate chain, further analytical tests, such as fluorescence spectrophotometry, were utilised to confirm such results.

#### 4.1.2. Confirmation of Fluorescence Labelling of Alginate with pH-Sensitive Fluorophore Using Fluorescence Signal Measurement

The chemical conjugation of the pH-sensitive fluorophores was also confirmed using the fluorescence measurement of each fluorophore-labelled alginate polymer after extensive dialysis to remove the unreacted fluorophore, as illustrated in Figure 7. It can be observed that the fluorescein-labelled alginate showed a fluorescence signal at 538 nm indicating the successful conjugation of the fluoresceinamine to the end of the alginate chain via the reductive amination process. Moreover, it can be concluded that there was a very weak conjugation between the Oregon green and the end of the alginate chain (OG-ALG (RA)), as indicated by its relatively lower fluorescence intensity amongst other fluorescently labelled alginate conjugates, demonstrating a low yield of alginate labelling with the OG via reductive amination. Therefore, other chemical conjugation strategies were performed to enhance the fluorescent labelling of the alginate with the OG, such as conjugating the OG to the intra-chain carboxylate groups of the alginate chain via a carbodiimide coupling reaction (EDC), which also demonstrated successful fluorescent labelling of the alginate. Moreover, the OG was additionally conjugated to the intra-chain aldehyde groups of two oxidised forms of alginate, and the emission of each fluorescently OG-labelled conjugates was measured. The alginate with 10% oxidation demonstrated a fluorescence signal similar to that of the EDC-coupled fluorescently labelled alginate, which reinforces the successful conjugation of the OG within the polymer chain aldehyde groups formed using periodate oxidation. On the other hand, the alginate with 80% oxidation with periodate showed a higher fluorescence intensity in comparison with the 10% oxidised alginate. This could be ascribed to the formation of a high number of aldehyde groups within the polymer chain, enabling a higher percentage of fluorophore conjugation. Overall, the fluorescence data measurements indicated successful alginate labelling with both pH-sensitive fluorophores, the Oregon green and the fluoresceinamine. Therefore, additional techniques, such as fluorescence spectroscopy, were used in order to confirm the successful attachment of the pH-sensitive fluorophores to the alginate polymers.

#### 4.1.3. Mean Hydrodynamic Diameter and Polydispersity Index (PDI) Measurements

The mean hydrodynamic diameters (MHDs) were measured for the different models of the pH-sensitive nanosensors prepared using different molar concentrations of calcium as a divalent cationic crosslinker, as demonstrated in Figure 8. It can be concluded that, with the various models of the pH-responsive nanosensors, the MHDs for the nanosensors generally increased as the concentration of the calcium increased, specifically at high Ca^+2^ concentrations (5 and 10 mM). Moreover, at the high calcium concentration of 10 mM, a significant increase in the particle size was observed with MHDs greater than 1000 nm, indicating the formation of aggregates of micrometre size. It should also be noteworthy that the end-chain conjugation of both pH-sensitive fluorophores to alginate-produced nanoparticles with relatively lower MHDs could be compared to the nanoparticles composed of the fluorescent alginate labelled with the OG using an intra-chain conjugation process such as the EDC coupling reaction, which was particularly observed at low calcium concentrations of 1.25 mM and 2.5 mM. This effect could be attributed to the bulky fluorophore (OG) that was covalently bound within the chain of the alginate. Consequently, increasing the steric hinderance between the alginate chains crosslinked by calcium resulted in the larger inner volumes and hydrodynamic diameters of the produced alginate nanoparticles. This effect was less pronounced with the oxidised forms of the alginate (10% and 80% oxidation), which could also be ascribed to the fact that, despite the fact that the OG was conjugated within the chains of the oxidised alginates, the oxidation process produced glycol split alginate chains with more flexible bonds. This could have led to the free rotation of the attached bulky fluorophore during alginate cross-linking and, thus, to the decreasing of the influence of fluorophore entrapment between the polymer chains and the resulting enlargement in the particle volume.

Additionally, the polydispersity index (PDI) values of the produced nanosensors were also measured and plotted for the different pH-responsive models, as shown in Figure 9. A correlation between the concentration of the crosslinker calcium and the PDI was found. Generally, the PDI significantly increased, reaching a value of around one (highly polydisperse) at the highest calcium concentration of 10 mM, in all the pH-responsive models irrespective of the process of fluorophore conjugation. This indicates that, at high calcium concentration, the produced pH-sensitive nanoparticles’ wide particles distribution could greatly influence the colloidal stability and the homogeneity of the nanosensors. Therefore, it was concluded that the lowest Ca^+2^ concentration should be used with a molar concentration of either 1.25 or 2.5 in order to the maintain maximum homogeneity of the produced pH-sensitive nanosensors with the lowest PDI values, i.e., below 0.5. Moreover, it could be observed that the lowest PDI value was achieved at the lowest crosslinker concentration with a pH nanosensors-based system consisting of the combined OG and fluorescein alginates with the fluorophore conjugated to the end of the polymer chain using the reductive amination method. This also reinforces that the end-chain conjugation of both pH-sensitive fluorophores showed a better particle size and particle size distribution when compared to the pH-sensitive models composed of the fluorescently labelled alginate prepared through the intra-chain conjugation of the pH-responsive fluorophores.

#### 4.1.4. Confirmation of the Nanosensors’ Charge Using Zeta Potential Measurements

Zeta potential values were also measured to further confirm the charges of the produced pH-responsive nanoparticles. Figure 10 shows the zeta potential values of the different models of pH-responsive nanosensors prepared at different concentrations of the cross-linker, i.e., calcium.

It can be observed that the zeta potential values were negative for all the pH-sensitive-nanosensors-based models; this is attributed to the presence of the carboxylate groups of the crosslinked alginate chains being exposed to the nanoparticles’ surfaces. Moreover, at a very low calcium concentration (1.25 mM), all the pH-responsive nanoparticles of all the prepared models displayed the highest negative zeta potential values (i.e., highest absolute values) compared to the pH nanosensors prepared using higher concentrations of the crosslinker Ca^+2^ ions. This could be explained by the lower proportion of divalent crosslinking of the negatively charged deprotonated carboxylate groups within the alginate polymer chains, consequently leaving a number of free/uncross-linked carboxylate groups exposed on the nanoparticle’s surface. It should also be noted that the pH nanosensors prepared through the end-chain attachment of both pH-sensitive fluorophores demonstrated the highest negative zeta potential values, specifically at a lower Ca^+2^ concertation (1.25 mM), which could indicate an anticipated high colloidal stability of this pH-responsive model due to the electrostatic repulsive forces, according to the DLVO theory. The high negative value of this model is thought to be due to the presence of higher free carboxylate groups, which are not covalently attached to the pH-sensitive fluorophore (OG), compared to the pH-sensitive system prepared through the intra-chain attachment of the fluorophore (OG) to the carboxylate groups of the alginate’s chain via EDC coupling. Surprisingly, the zeta potential values for the pH nanosensors prepared through the intra-chain conjugation of the fluorophore (OG) to the formed aldehyde groups within the chain of oxidised alginate (10% and 80% oxidation), which should not presumably influence the free carboxylate groups within the alginate polymer chains; however, the engagement of these carboxylate groups through calcium crosslinking could be a possible explanation for the unexpected lower zeta potential values for these pH-responsive models.

### 4.2. Calibration of the Designed pH Nanosensors Using pH versus Fluorescence Signal Response Curve

The emission at 538 nm was plotted against the pH of different models of the pH-responsive nanosensors prepared using different concentrations of the calcium crosslinker. Figure 11 shows the calibration curves for the following: the produced pH-sensitive nanosensors, which consist of fluorescently labelled alginate polymers prepared through the end-chain conjugation of both fluorophores, fluoresceinamine and Oregon green, using the reductive amination method, i.e., equal ratios of fluorescein-labelled alginate and Oregon green-labelled alginate (A); the pH nanosensors prepared through the end-chain conjugation of the fluoresceinamine whilst conjugating the Oregon green within the polymer chain using the EDC coupling reaction, i.e., equal ratios of fluorescein-labelled alginate and alginate labelled with Oregon green coupled using EDC (B); the pH nanosensors consisting of equal ratios of two fluorescently labelled alginate polymers prepared through the end-chain conjugation of the fluoresceinamine whilst conjugating the Oregon green within the polymer chain after 10% of alginate oxidation with periodate (C); or the pH nanosensors composed of equal ratios of two fluorescently labelled alginate polymers prepared through the end-chain conjugation of the fluoresceinamine whilst conjugating the Oregon green within the polymer chain after 80% of alginate oxidation with periodate (D). A correlation between the nanosensors’ fluorescence signal measured at 530 nm and the pH values was generated for each pH-responsive model. The data demonstrated that all the pH-responsive systems followed a sigmoidal model of curve fitting similar to that reported in the literature [20], with a relatively high correlation, i.e., a high coefficient of determination (R^2^). Generally, it should be noted that all the pH nanosensors displayed an increase in the fluorescence intensity with respect to the increase in the pH values, reaching the maximum fluorescence value (near plateau) at the highest pH values of 7.5 to 8. This indicates the successful pH-responsiveness of the developed models of pH nanosensors. The pH-sensitivity of the developed nanosensors could be attributed to the presence of both pH-sensitive fluorophores incorporated within the matrix of the nanoparticles—namely, the Oregon green, which responds to the pH in the range of 3.5 to 5, and the fluoresceinamine, which responds to the pH values ranging from 5 to 7.5. This strategy of combining two different pH-sensitive fluorophores allowed us to produce pH nanosensors with an extended dynamic range, i.e., 3.5–7.5. It was also observed that all the models of the pH nanosensors demonstrated a relatively higher fluorescence intensity at the highest calcium concentration (10 mM). This could be ascribed to the larger particles’ diameters/volumes produced using a high crosslinker concentration, and, thus, slightly higher amounts of the fluorophore-conjugated alginate chains were incorporated within the matrix of the nanoparticles. However, the physical characteristics of the formed particles using a high calcium concentration (10 mM), such as large mean hydrodynamic diameters (aggregates of micrometre size) and high PDI values, limit their use for nano-biosensing applications.

On the other hand, the pH nanosensors prepared using the lowest crosslinker concentration showed the lowest fluorescence intensity compared to the other pH nanosensors that had been prepared using higher concentrations of the crosslinker. This could also be attributed to the lowest average particle size formed at the low calcium concentration, meaning that, consequently, the minimum amounts of the fluorescently labelled alginates were crosslinked within the particles’ matrices. Moreover, the pH nanosensors prepared by either combining the fluorescein-labelled alginate (end-chain) with the Oregon green-labelled alginate (end-chain), i.e., model I, or combining the fluorescein-labelled alginate (end-chain) with the Oregon green-labelled alginate (intra-chain via EDC coupling), i.e., model II, demonstrated good pH-response correlations when prepared using a low crosslinker concentration (1.25 mM), with R^2^ values of 0.9916 and 0.9879, respectively. In contrast, the pH-nanosensors prepared by either combining the fluorescein-labelled alginate (end-chain) with the Oregon green-labelled alginate (intra-chain of 10% oxidised alginate), i.e., model III, or combining the fluorescein-labelled alginate (end-chain) with the Oregon green-labelled alginate (intra-chain of 80% oxidised alginate), i.e., model IV, both showed relatively poor pH-response correlations when prepared using a low calcium concentration (1.25 mM), with R^2^ values of 0.8872 and 0.9261, respectively. This is thought to be due to a poor cross-linking of the alginate chains or an uneven distribution of the fluorophore-labelled alginate within the nanoparticles’ matrices, which could be attributed to the aldehyde groups’ formation within the polymer chain after the alginate’s oxidation, followed by the fluorophore’s conjugation, which might have sterically hindered the alginate’s crosslinking using calcium.

It should also be noted that model II of the pH nanosensors prepared using a cross-linker concentration of 5 and 10 mM, respectively, displayed the highest pH-response correlation amongst the other pH-responsive models, with R^2^ values of 0.9980. However, the pH-sensitive particles of this model prepared using 10 mM of calcium demonstrated large hydrodynamic diameters and high PDI values, indicating that they are not suitable for biosensing applications such as intracellular pH measurements.

To sum up, there is an optimum concentration of the divalent crosslinker calcium, as well as an optimum fluorophore conjugation method of the utilised fluorescently labelled alginate for producing pH-sensitive fluorescently labelled alginate nanoparticles with the maximum pH-responsiveness and the highest pH-fluorescence intensity correlation.

## 5. Conclusions

A new analytical approach has been adopted using a clean synthesis of pH nanosensors consisting of pH-sensitive fluorophores covalently attached to an alginate polysaccharide chain. The fluorophores were covalently linked to the alginate polymer chains using various chemical conjugation methods to produce pH nanosensors with an extended dynamic range that can detect the pH changes from 3.5 to 7.5. These fluorophores encompass Oregon green, which is sensitive to pH ranging from 3.5 to 5, and fluoresceinamine, which is sensitive to pH values ranging from 5 to 7.5. The conjugation of the pH-sensitive fluorophores to the alginate chain was confirmed using FTIR spectra and fluorescence signal measurements.

Various pH-responsive models were prepared by co-crosslinking a fluorescein-labelled alginate and an Oregon green-labelled alginate with a divalent cation (Ca^+2^) using microfluidic technology to produce the pH-sensitive alginate nanoparticles. Model I consisted of a combination of fluoresceinamine-labelled alginate and Oregon green-labelled alginates, where both fluorophores were conjugated to the end of alginate chain using the reductive amination process. Instead, model II comprised a combination of fluoresceinamine- and Oregon green-labelled alginates, where the fluoresceinamine was conjugated to the end of the alginate chain using reductive amination, and the Oregon green was conjugated within the alginate chain using the EDC coupling method. In addition, model III contained a combination of fluoresceinamine- and the Oregon green-labelled alginates where the fluoresceinamine was conjugated to the end of the alginate chain using reductive amination, and the Oregon green attached within the chain of aldehyde groups of the oxidised alginates (at 10% oxidation level) using the reductive amination method, whilst model IV consisted of a combination of fluoresceinamine- and Oregon green-labelled alginates, where the fluoresceinamine was conjugated to the end of the alginate chain using reductive amination and the Oregon green attached within the chain of aldehyde groups of the oxidised alginates (at 80% oxidation level) using the reductive amination method.

All the models of the pH-responsive nanosensors were characterised according to their mean hydrodynamic diameter (MHD), polydispersity index (PDI), and zeta potential values using the dynamic light scattering technique (DLS). The data showed that the nanosensors demonstrated smaller mean hydrodynamic diameters and lower PDI values when a low concentration of the crosslinker (Ca^+2^) was used. Also, all the pH nanosensors displayed negative zeta potential values, indicating the ionisation of the carboxylate groups of the alginate chains on the particles’ surfaces, which reinforced the colloidal stability of the nanosensors.

All the pH-responsive models demonstrated a successful pH-fluorescence response with a sigmoidal increase in the fluorescence intensity as the pH values raised. Apart from the model III and the model IV of the pH-responsive nanosensors that were prepared at a low crosslinker concentration (1.25 mM Ca^+2^), all the other pH-responsive models displayed sigmoidal pH-fluorescence response curves with high coefficient of determination values (R^2^), indicating a high degree of correlation between the detected pH values and the measured fluorescence signal. Although the pH nanosensors prepared at a high concentration of the cross-linker (10 mM) showed a relatively high pH-responsive value (high R^2^ value), they showed physical properties that would hinder their applications in pH biosensing, such as in intracellular pH measurements. The data showed that the low MHDs with relatively acceptable PDIs of the produced pH nanosensors were achieved at the lowest concentration of the cross-linker (Ca^+2^), 1.25 mM, suggesting the utilisation of these pH nanosensors for further bio-sensing applications.

In conclusion, we have unveiled that various factors may play a role in developing pH-responsive alginate-based nanosensors, including the concentration of the crosslinker, the method of conjugating the pH-sensitive fluorophores to the alginate chain, and the concentration of the fluorescently labelled alginate polymer. To this end, we strongly recommend using alginate-based pH nanosensors as potential alternative diagnostics for the pH detection of biological systems. The developed pH-responsive system leverages new advantages in comparison to the conventional pH nanosensors, including the ecologically friendly approach of synthesis which completely precludes the use of organic solvents, thus minimising their potential harm to the environment. Moreover, the designed alginate-based pH nanosensors hold numerous potential diagnostics and biosensing insights for the future. For instance, alginate-based pH nanosensors could be harnessed for early diagnosis of diseases that are characterised by a low luminal pH due to the high secretion of protons, such as gastric carcinoma and Zollinger Ellison syndrome. Moreover, the developed pH nanosensors could be utilised for the detection of intracellular pH, which could be lower than the physiological pH in some diseases such as various types of carcinomas, which could enable early cancer detection and could enhance the treatment outcomes at early stages. Additionally, the pH-responsive alginate nanosensors could be employed in monitoring the pharmacological activity/efficiency of some drugs that interfere with the acid production mechanism in the gastric lumen, including proton pump inhibitors (e.g., omeprazole) and antacids, which could enhance the prognosis of diseases characterised by high gastric acid secretion, such as peptic ulcer.

## Figures and Tables

**Figure 1 sensors-23-08453-f001:**
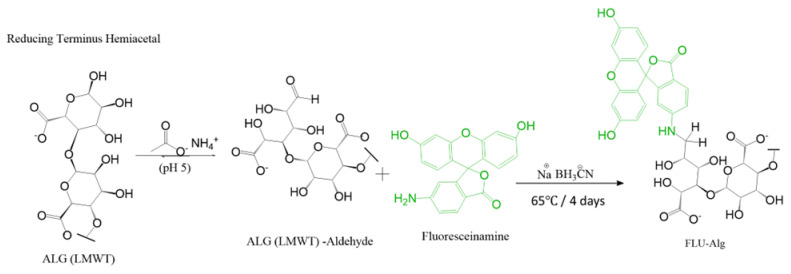
Reaction scheme presenting the end-chain reductive amination reaction between alginate and fluoresceinamine (FLU-ALG).

**Figure 2 sensors-23-08453-f002:**
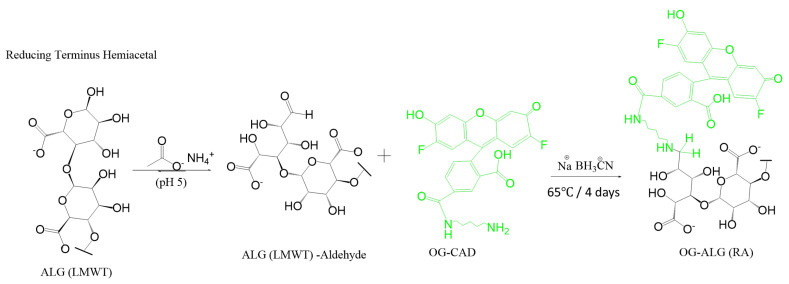
Reaction scheme presenting the end-chain reductive amination reaction between alginate and Oregon green (OG-ALG (RA)).

**Figure 3 sensors-23-08453-f003:**
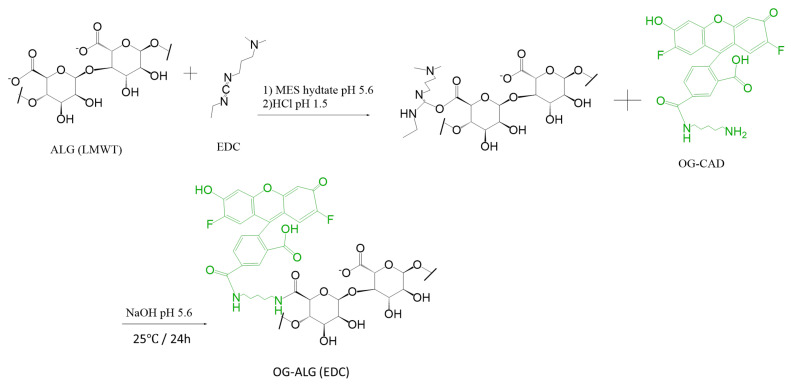
Reaction scheme presenting the intra-chain, EDC coupling reaction between alginate and Oregon green (OG-ALG (EDC)).

**Figure 4 sensors-23-08453-f004:**
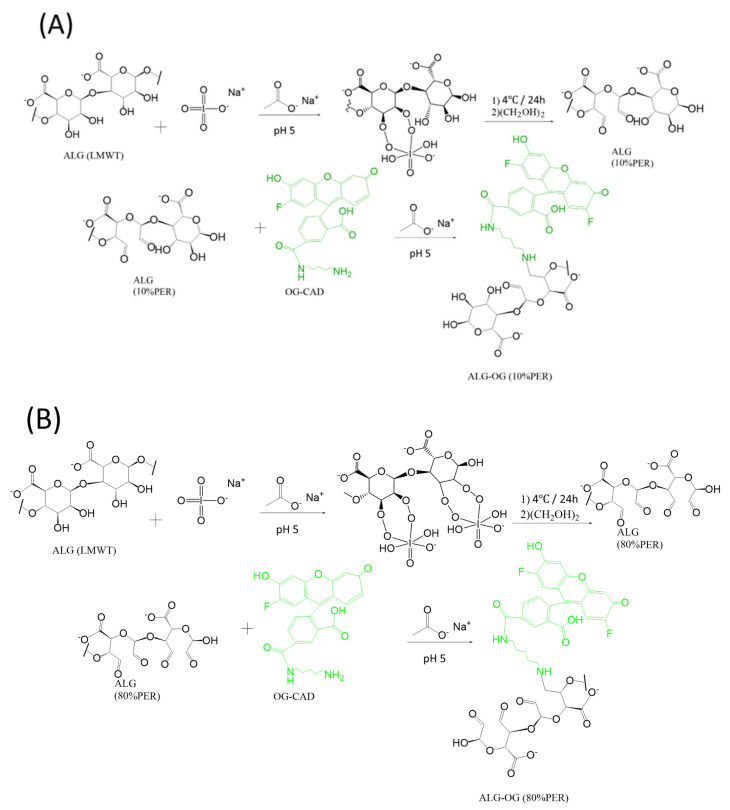
Reaction scheme presenting the oxidative intra-chain reductive amination reaction between Oregon green and two forms of oxidized alginate chains with 10% (**A**) and 80% (**B**) oxidation percentages, providing (ALG-OG (10%PER)) and (ALG-OG (80%PER)), respectively.

**Figure 5 sensors-23-08453-f005:**
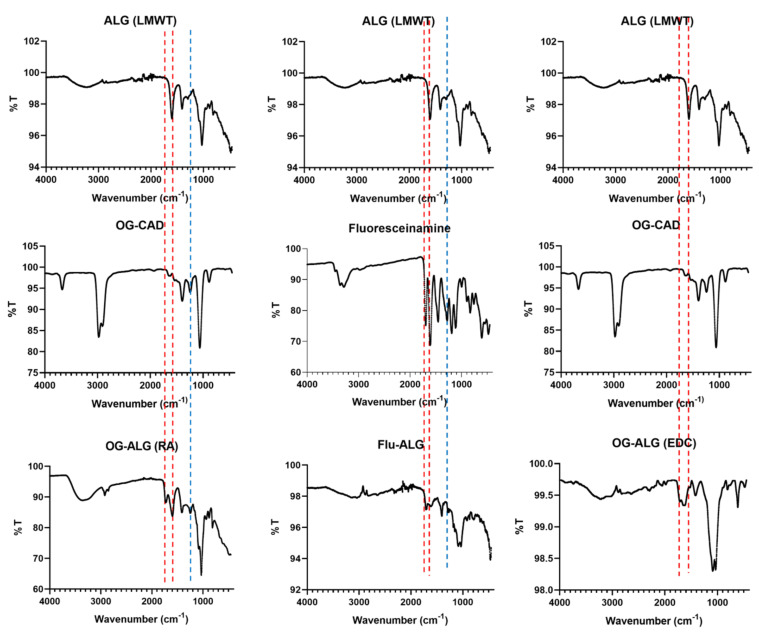
FTIR spectra representing the transmission peaks plotted against the wavenumber (cm^−1^) of low molecular weight alginate polymer (ALG-LMWT), fluoresceinamine, Oregon green cadaverine (OG-CAD), fluoresceinamine-labelled alginate (FLU-ALG), and Oregon green-labelled alginate using either direct reductive amination (OG-ALG (RA)) or EDC coupling reaction (OG-ALG (EDC)).

**Figure 6 sensors-23-08453-f006:**
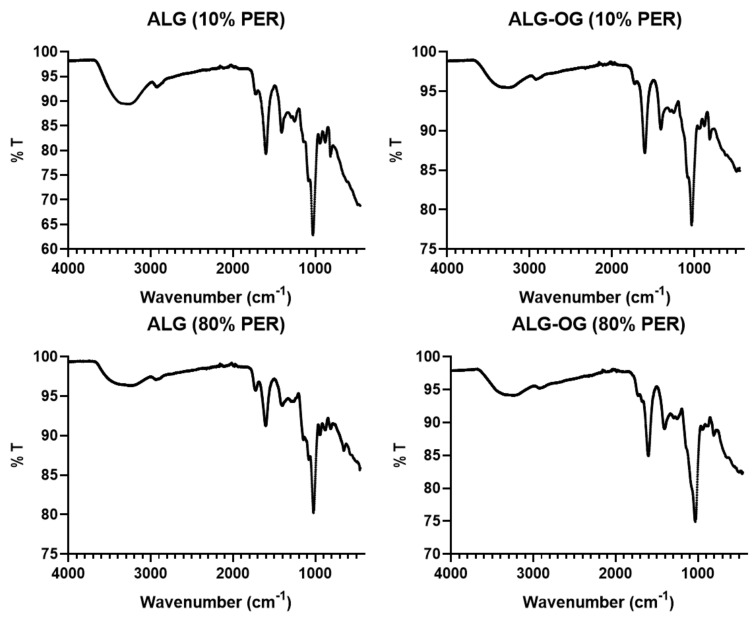
FTIR spectra representing the transmission peaks plotted against the wavenumber (cm^−1^) of low molecular weight alginate polymer after chemical modification (oxidation) using periodate to produce aldehyde groups with either 10% molar ratio of periodate to alginate (ALG (10% PER)) or 80% molar ratio of periodate to alginate (ALG (80% PER)), and the same polymers after chemical conjugation with Oregon green using the reductive amination process.

**Figure 7 sensors-23-08453-f007:**
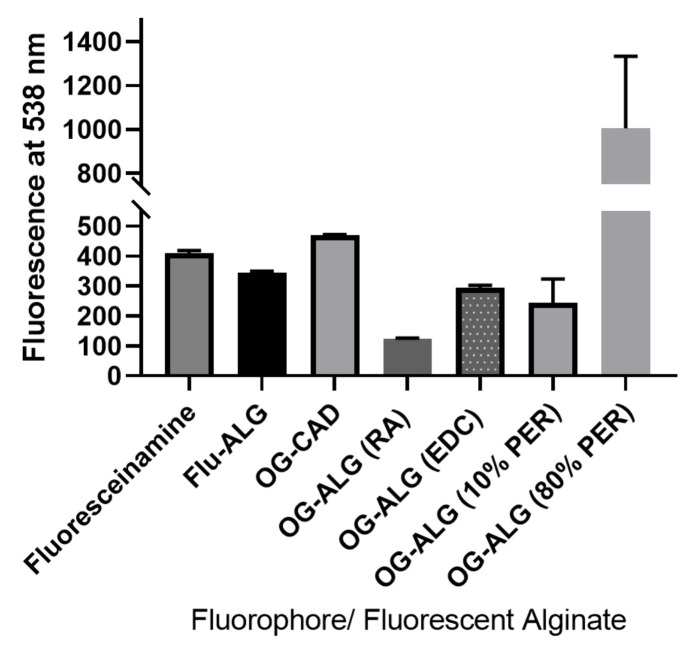
Fluorescence measurements at an emission wavelength of 538 nm for fluoresceinamine, Oregon green cadaverine (OG-CAD), fluoresceinamine-labelled alginate (FLU-ALG), and Oregon green-labelled alginate using either direct reductive amination (OG-ALG (RA)) or EDC coupling reaction (OG-ALG (EDC)), as well as Oregon green-labelled oxidised alginate prepared by either 10% molar ratio of periodate to alginate (ALG (10% PER)) or 80% % molar ratio of periodate to alginate (ALG (80% PER)).

**Figure 8 sensors-23-08453-f008:**
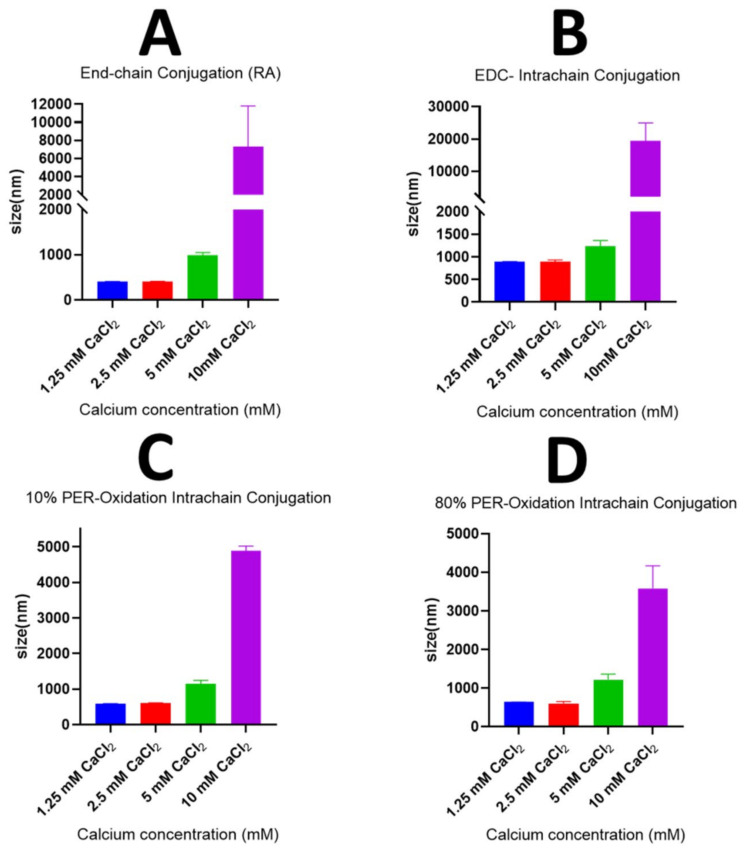
Graphs illustrating the mean particle size (nm) against the calcium concentration (mM) of the pH nanosensors composed of fluorescently labelled alginate polymers prepared using the following methods: the end-chain conjugation of both fluorophores, fluoresceinamine and Oregon green, using the reductive amination method (**A**); the end-chain conjugation of the fluoresceinamine whilst conjugating the Oregon green within the polymer chain using the EDC coupling reaction (**B**); the end-chain conjugation of the fluoresceinamine whilst conjugating the Oregon green within the polymer chain after 10% of alginate oxidation with periodate (**C**); or the end-chain conjugation of the fluoresceinamine whilst conjugating the Oregon green within the polymer chain after 80% of alginate oxidation with periodate (**D**).

**Figure 9 sensors-23-08453-f009:**
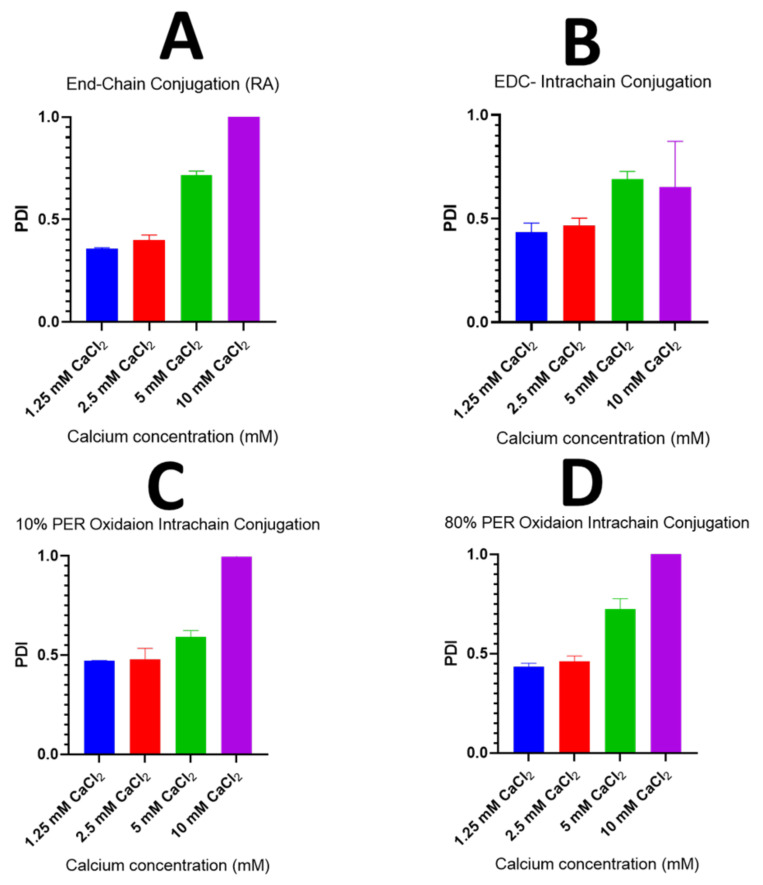
Graphs demonstrating the polydispersity index (PDI) of the produced pH nanosensors of different pH-responsive models consisting of fluorescently labelled alginate polymer prepared using one of the following methods: the end-chain conjugation of both fluorophores, fluoresceinamine and Oregon green, using the reductive amination method (**A**); the end-chain conjugation of the fluoresceinamine whilst conjugating the Oregon green within the polymer chain using the EDC coupling reaction (**B**); the end-chain conjugation of the fluoresceinamine whilst conjugating the Oregon green within the polymer chain after 10% of alginate oxidation with periodate (**C**); or the end-chain conjugation of the fluoresceinamine whilst conjugating the Oregon green within the polymer chain after 80% of alginate oxidation with periodate (**D**).

**Figure 10 sensors-23-08453-f010:**
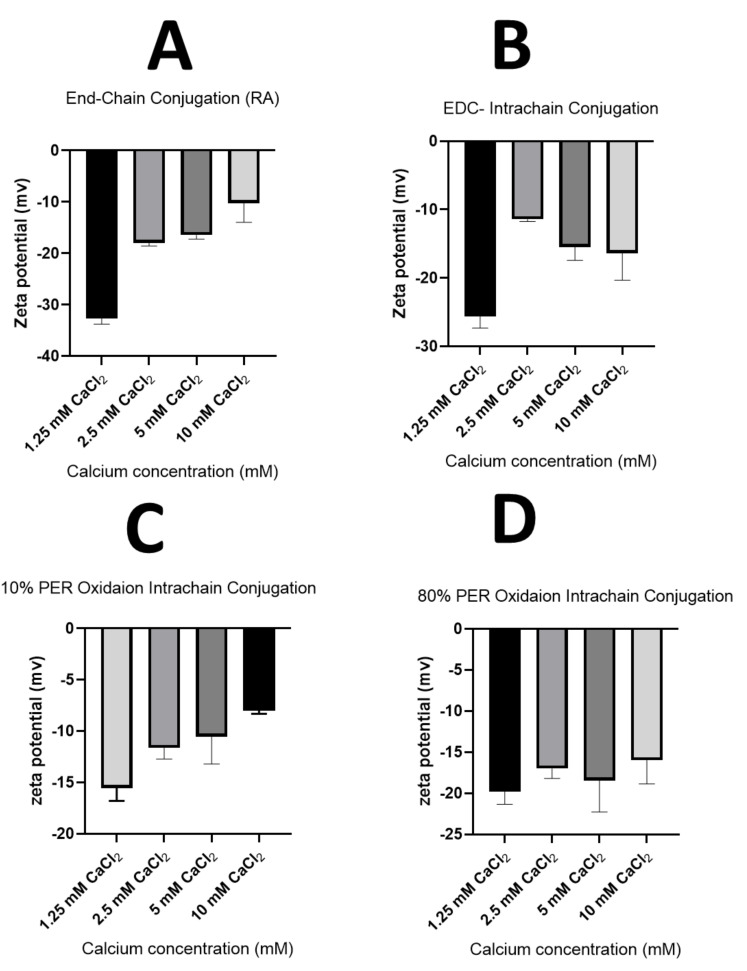
Graphs showing the zeta potential values against the calcium concentration (mM) of the pH nanosensors composed of fluorescently labelled alginate polymers prepared using the following methods: the end-chain conjugation of both fluorophores, fluoresceinamine and Oregon green, using the reductive amination method (**A**); the end-chain conjugation of the fluoresceinamine whilst conjugating the Oregon green within the polymer chain using the EDC coupling reaction (**B**); the end-chain conjugation of the fluoresceinamine whilst conjugating the Oregon green within the polymer chain after 10% of alginate oxidation with periodate (**C**); or the end-chain conjugation of the fluoresceinamine whilst conjugating the Oregon green within the polymer chain after 80% of alginate oxidation with periodate (**D**).

**Figure 11 sensors-23-08453-f011:**
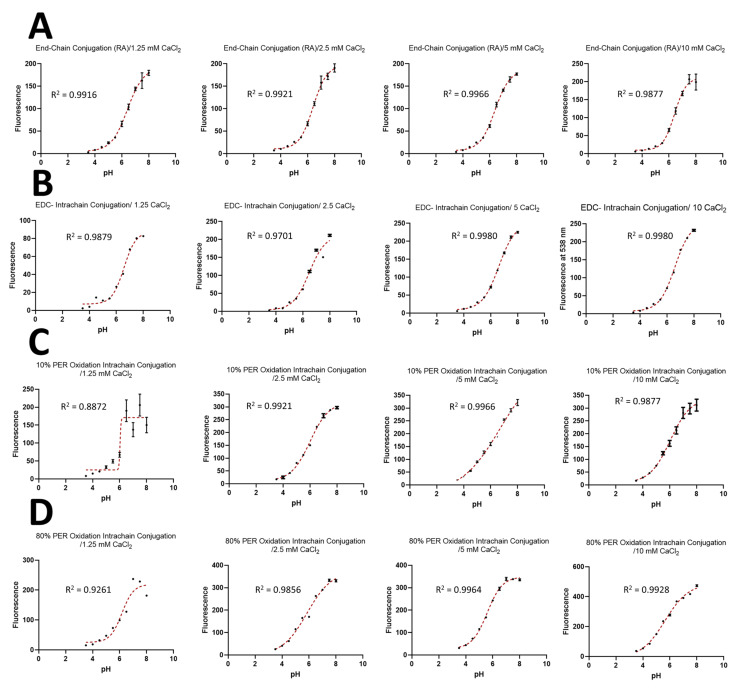
Diagrams representing the fluorescence response with respect to pH of the various models of pH-responsive nanosensors, prepared using different calcium concentrations, composed of fluorescently labelled alginate polymers prepared using one of the following methods: the end-chain conjugation of both fluorophores, fluoresceinamine and Oregon green, using the reductive amination method (**A**); the end-chain conjugation of the fluoresceinamine whilst conjugating the Oregon green within the polymer chain using the EDC coupling reaction (**B**); the end-chain conjugation of the fluoresceinamine whilst conjugating the Oregon green within the polymer chain after 10% of alginate oxidation with periodate (**C**); or the end-chain conjugation of the fluoresceinamine whilst conjugating the Oregon green within the polymer chain after 80% of alginate oxidation with periodate (**D**).

**Table 1 sensors-23-08453-t001:** The most prominent FTIT absorption peaks measured for either dye, polymer, or polymer–dye conjugates as follows: alginate polymer (ALG-LMWT), fluoresceinamine, Oregon green cadaverine (OG-CAD), fluoresceinamine-labelled alginate (FLU-ALG) using the reduction amination process, and Oregon green-labelled alginate using either direct reductive amination (OG-ALG (RA)) or EDC coupling reaction (OG-ALG (EDC)), as well as low molecular weight alginate polymers after chemical modification (oxidation) using periodate to produce aldehyde groups with either 10% molar ratio of periodate to alginate (ALG (10% PER)) or 80% molar ratio of periodate to alginate (ALG (80% PER)), and the same polymers after chemical conjugation with Oregon green using the reductive amination process.

	Functional Group	O-H Stretching	C-H Stretching	C=O Stretching (Carboxylate)	C=O Stretching (Aldehyde)	C=O Stretching (Lactone /Cyclohexanone) C=C Stretching	C=O Stretching (Amide)	O-H Bending C-H Bending	C-N Stretching
Polymer, Dye or Polymer–Dye Conjugate									
**ALG(LMWT)**		3320–3120	2920	1600	-	-	-	1405	-
**Og-CAD**		3671	2975	1638	-	1640	1595	1394	1243 (**or C-F** **stretching**)
**Fluoresceinamine**		3100–3600	2860	1608	-	1696	-	1457	1281
**Flu-ALG (RA)**		3437–3181	2920	1598	-	1731	-	1410	1240
**OG-ALG (EDC)**		3540–2925	2920	1600	-	1645	1705	1413	1160 (**or C-F** **stretching**)
**ALG (10% PER)**		3400–3190	2920	1599	1722	-	-	1402	-
**ALG (80% PER)**		3400–3100	2920	1602	1726	-	-	1402	-
**ALG-Og (10% PER)**		3400–3190	2920	1602	1722	-	-	1406	1240 (**or C-F** **stretching**)
**ALG-Og (80% PER)**		3426–3160	2920	1602	1726	-	-	1406	1251 (**or C-F** **stretching**)

## Data Availability

All data are available in the manuscript.

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
