# Peer review of "Development of a Novel, Ecologically Friendly Generation of pH-Responsive Alginate Nanosensors: Synthesis, Calibration, and Characterisation"

_sensors, 2023, doi:10.3390/s23208453_

Round 1

Reviewer 1 Report

While it is an interesting paper, the authors need to address the following issues:
1) In the paragraph starting at line 109, the authors mention some examples of nanosensors such as silica nanoparticles that require ethanol for cleaning and polyacrylamide requiring hexane as a solvent. The dangers to the environment from the use of these chemicals is not clear.

2) In line 119, the authors declare that these chemical have a detrimental effect on the environment but fail to provide reasons or cite some literature to support their argument. 

3) Please improve the write-up of sections 2.2.2, 2.2.3, and 3.1. They are very confusingly written making it hard to understand. 

4) Excessive self-citation is not a good academic practice, please improve on that as well. 

Author Response

While it is an interesting paper, the authors need to address the following issues:
1) In the paragraph starting at line 109, the authors mention some examples of nanosensors such as silica nanoparticles that require ethanol for cleaning and polyacrylamide requiring hexane as a solvent. The dangers to the environment from the use of these chemicals is not clear.

Thanks very much for the comment. The danger of using these organic solvents was addressed in the text  and proper citations were included.

2) In line 119, the authors declare that these chemicals have a detrimental effect on the environment but fail to provide reasons or cite some literature to support their argument. 

Thank you very much for the feedback. The harmful effects of these chemicals were backed by the literature and reflected on the text.

3) Please improve the write-up of sections 2.2.2, 2.2.3, and 3.1. They are very confusingly written making it hard to understand. 

Thanks for your constructive input. All indicated sections were re-written as suggested.

4) Excessive self-citation is not a good academic practice, please improve on that as well. 

Thanks so much for the valuable comment. Self-citations were removed from most of the article sentences and replaced by non-self-relevant research work conduced by other researchers.

Reviewer 2 Report

The manuscript entitled " Development of a Novel Ecological-Friendly Generation of pH-Responsive Alginate Nanosensors: Synthesis, Calibration, and Characterization" introduced designing alginate-based pH-responsive nanosensors without requiring organic solvent for synthesis and are biocompatible, biodegradable, and environmental-friendly. They synthesized different models of the pH-responsive nanoparticles by varying the method of fluorophore conjugation using an ecological-friendly synthetic approach and characterized them with various methods. The article is organized well, and it is structured in a good shape. However, there are some issues that should be addressed prior to possible publication. The following points are strongly suggested to be addressed:

-          Abstracts should contain the most key parts and important results and it should be short/informative/attractive. Please revise the abstract.

-          The introduction section is too wordy. It is suggested to improve the structure of the introduction by summarizing some of the information and presenting 3 cohesive paragraphs with the first two emphasizing the importance of the work/literature and the last one describing a gist of the work. Also, the importance of biosensors in providing accurate detection of various biomarkers and various nanomaterials/structures application should be addressed. Some potential papers could be beneficial (https://doi.org/10.1016/j.tibtech.2023.04.001), (https://doi.org/10.1016/j.tibtech.2023.04.001), (https://doi.org/10.1002/adfm.202010388), 

-          It is suggested to proofread the article for small typos and/or inconsistent styling.

-          In figures (8-11) the label of subfigures seems too large compared to other parts. Please revise them appropriately.

-          Please rewrite the conclusion part in a format that can attract the reader. Please provide insights for future prospective implementation of designed nanosensors.

          It is suggested to proofread the article for small typos and/or inconsistent styling.

Author Response

The manuscript entitled " Development of a Novel Ecological-Friendly Generation of pH-Responsive Alginate Nanosensors: Synthesis, Calibration, and Characterization" introduced designing alginate-based pH-responsive nanosensors without requiring organic solvent for synthesis and are biocompatible, biodegradable, and environmental-friendly. They synthesized different models of the pH-responsive nanoparticles by varying the method of fluorophore conjugation using an ecological-friendly synthetic approach and characterized them with various methods. The article is organized well, and it is structured in a good shape. However, there are some issues that should be addressed prior to possible publication. The following points are strongly suggested to be addressed:

-          Abstracts should contain the most key parts and important results and it should be short/informative/attractive. Please revise the abstract.

Thank you very much for the feedback. Abstract was revised and rephrased as recommended; changes are highlighted.

-          The introduction section is too wordy. It is suggested to improve the structure of the introduction by summarizing some of the information and presenting 3 cohesive paragraphs with the first two emphasizing the importance of the work/literature and the last one describing a gist of the work. Also, the importance of biosensors in providing accurate detection of various biomarkers and various nanomaterials/structures application should be addressed. Some potential papers could be beneficial (https://doi.org/10.1016/j.tibtech.2023.04.001), (https://doi.org/10.1016/j.tibtech.2023.04.001), (https://doi.org/10.1002/adfm.202010388), 

Thanks for the valuable comment. Introduction was revised as suggested, re-written in a more concise manner and fewer number of words yet retaining same message. Also, all suggested papers were  properly cited in the text as suggested.

-          It is suggested to proofread the article for small typos and/or inconsistent styling.

Thanks very much for the constructive feedback. Article was spell-checked for typos and corrected.

-          In figures (8-11) the label of subfigures seems too large compared to other parts. Please revise them appropriately.

Thanks so much for the comment.  Legends of the subfigures were all adjusted to be consistently at font size 12 and type Arial.

-          Please rewrite the conclusion part in a format that can attract the reader. Please provide insights for future prospective implementation of designed nanosensors.

Thank you for the valuable feedback. Conclusion was re-written as suggested and future insights for the potential applicants of the designed pH-nanosensors were included and highlighted in yellow.

Comments on the Quality of English Language

          It is suggested to proofread the article for small typos and/or inconsistent styling.

Thanks very much. Article was spell-checked and typos were corrected. Also, sentences with inconsistent styling were re-phrased.

Reviewer 3 Report

 Alwraikat et al. presented an alginate base nanoparticle with pH responsiveness. Overall, the work is interesting. Some edits can be made to improve the overall quality:

1. Line 116-117, check grammar: "...e.g., CdSe QDs also requires 116 etc. chloroform..."

2. Lines 119-130, does not fit into context. Consider removing it from the text

3. Nanoparticles should be characterizable in EM. Please provide some physical characterization of the synthesized particles.

4. Section 3.1 and Figure 5 appear to be connected. Please consider combining or reworking the section

5. Section 3.2, please show results (i.e. spectrum)

6. Overall, 3.1-3.3 appears to convey similar information in terms of characterization. They can be combined together and split into smaller sub-sections later on.

7. In Figure 11, the data points are smaller than the fit. Also, consider consolidating the pH response of different ion concentrations into one graph. 

8. The author states their synthesized pH-responsive nanoparticle is biocompatible. Please consider supporting that claim with evidence. 

minor edits, and format revision

Author Response

 Alwraikat et al. presented an alginate base nanoparticle with pH responsiveness. Overall, the work is interesting. Some edits can be made to improve the overall quality:

  1. Line 116-117, check grammar: "...e.g., CdSe QDs also requires 116 etc. chloroform..."

Thanks for the comment. Grammatical error was corrected as suggested, it refers to the synthetic procedure which is singular.

  1. Lines 119-130, does not fit into context. Consider removing it from the text

Thanks very much for the constructive feedback. Text was removed as suggested.

  1. Nanoparticles should be characterizable in EM. Please provide some physical characterization of the synthesized particles.

Thank you very much for the feedback. Electron Microscopy (EM) is a very powerful tool for characterising the synthesised nanoparticles. But unfortunately, we do  not have EM facility at our university in a low-middle income facilities and we hope to improv this in the future. Moreover, results from DLS could be reliable in this case as the major focus of this paper is the pH responsive of the nanoparticles rather than the six itself.

  1. Section 3.1 and Figure 5 appear to be connected. Please consider combining or reworking the section

Thanks very much for the valuable comment. Both figure 5 and figure 6 must come under section 3.1 and subsection 3.1.1 i.e., “Confirmation of fluorescence labelling of alginate with pH sensitive fluorophore by FTIR” as indicated in the MS Word version of our submitted manuscript. Probably an error has occurred during the journal/publisher PDF generation

  1. Section 3.2, please show results (i.e. spectrum)

Thanks so much for the valuable comment. End-point measurements were included in our manuscript’s MS Word version but not sure why they did not appear in the MDPI-PDF version (maybe technical error). Moreover, determining endpoints fluorescence signal we believe is sufficient in this case as the emission of fluorescein dye is well reported in the literature emission 525 nm, however the closest emission filter installed our instrument (FLx 800 Microplate fluorescence  Reader -BioTek) was 538 which could collect 84% of the emitted photons as indicated by ThermoFisher Spectra Viewer:  https://www.thermofisher.com/order/fluorescence-spectraviewer/#!/.

  1. Overall, 3.1-3.3 appears to convey similar information in terms of characterization. They can be combined together and split into smaller sub-sections later on.

Sections were merged and sub-sections were created as suggested.

  1. In Figure 11, the data points are smaller than the fit. Also, consider consolidating the pH response of different ion concentrations into one graph. 

Thanks very much for the comment. Figures were re-drawn but could not increase the size of the data points as all error bars have disappeared when I tried to do so however I tried to make the fit dashed bar thinner. Also, we have tried to consolidate the pH response of different ion concentrations for various models into one graph and we obtained a very cumbersome graph that is quite difficult for the reader to grasp and understand that is particularly because many datasets/points overlap together. Therefore, we strongly recommend keeping them in separate graphs unless strongly advised/demanded by the reviewer.

  1. The author states their synthesized pH-responsive nanoparticle is biocompatible. Please consider supporting that claim with evidence.

I appreciate your feedback. The compatibility of alginate nanoparticles was backed by the literature and citations were included in the introduction; amendments were highlighted in green.

Round 2

Reviewer 1 Report

I think the paper has been fairly improved and can be considered for publication

Reviewer 2 Report

The revised version addressed all the comments.